# Chordal Graph Sampling-Based Mini-batch Training Algorithm for Large Graphs

## Abstract

Graph Neural Networks (GNNs) are powerful models for learning representations of attributed graphs. To scale GNNs to large graphs, many methods use various techniques, such as sampling and decoupling, to alleviate the "neighbor explosion" problem during mini-batch training. However, these sampling-based mini-batch training methods often suffer from greater information loss than decoupling-based methods or full-batch GNNs. Besides, most original segmentation methods for large graphs usually lose a large number of edges, resulting in suboptimal performance when performing mini-batch training. Therefore, we propose a **C**hordal **G**raph **S**ampling-based mini-batch **T**raining algorithm for GNNs on large scale graph datasets, called **CGST**. CGST includes a balanced chordal graph partition module and a batch random aggregation module to improve performance on node classification tasks while maintaining main information of the original graph structure. Experiments on three large-scale graph datasets prove the effectiveness of CGST.

## 1 Introduction

The Graph Neural Network(GNN) has garnered significant attention in the realm of graph-based applications, encompassing tasks such as semi-supervised node classification (Kipf & Welling, 2017), link prediction (Zhang & Chen, 2018), and recommender systems (Ying et al., 2018a). Within the framework of a given graph, GNN employs graph convolutional operations to compute node embeddings across multiple layers iteratively. At each layer, a node's embedding is derived by aggregating information from its neighboring nodes, subsequently undergoing one or more layers of linear transformations and nonlinear activations. The resultant embedding at the final layer is then utilized for various downstream tasks. For instance, in scenarios involving node classification, the final layer embedding is fed into a classifier to infer node labels, facilitating end-to-end training of GNN parameters.

Despite the notable achievements of GNNs in numerous graph-related applications, the training of GNNs for extensive graphs poses a significant challenge. Unlike text or images with usually constrained lengths or sizes, practical graph data can frequently present itself in immensely expansive scales or dimensions, reflecting the complexity and magnitude inherent in real-world graph structures. For instance, the 2019 Facebook social network comprises 2.7 billion users (Leskovec & Mcauley, 2012), exemplifying the immense scale of real-world graphs. *Managing such large-scale graphs through full-batch GNN training, where all nodes are processed together to update parameters, is infeasible.* Nevertheless, previous mini-batch GNN training methods often suffer from huge information loss, since the connections and cross-community nodes are removed when partitioning the large-scale graphs into subgraphs. Consequently, the training of deep and expansive GNNs remains a formidable task, impeding their deployment in various large-scale graph applications, including social networks, recommender systems, and knowledge graphs.

To solve the issues, researchers have proposed sampling-based methods to train GNNs based on mini-batch of nodes, which only aggregate the embeddings of a sampled subset of neighbors of each node in the mini-batch. Among them, one direction is to use a node-wise neighbor-sampling method. For example, GraphSAGE (Hamilton et al., 2017) calculates each node embedding by leveraging only a fixed number of uniformly sampled neighbors. Although this kind of approach reduces the computation cost in each aggregation operation, the total cost can still be large. As

pointed out in (Hardt et al., 2016), the recursive nature of node-wise sampling brings in redundancy for calculating embeddings. Even if two nodes share the same sampled neighbor, the embedding of this neighbor has to be calculated twice.

Such redundant calculation will be exaggerated exponentially when the number of layers increases. Following this line of research as well as reducing the computation redundancy, a series of work was proposed to reduce the size of sampled neighbors. VR-GNN (Chen et al., 2017) proposes to leverage variance reduction techniques to improve the sample complexity. Cluster-GCN (Chiang et al., 2019) considers restricting the sampled neighbors within some dense subgraphs, which are identified by a graph clustering algorithm before the training of GNN. However, these methods still cannot well address the issue of information loss when simplifying training procedures or model architectures, which may become worse when training very deep and large GNNs.

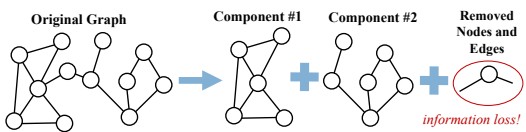

Figure 1: The diagram of former subgraph-sampling based methods. Previous mini-batch training methods often suffer from information loss due to the removed nodes and edges.

In this paper, we propose a novel subgraph-sampling based mini-batch training method called CGST to efficiently train GNNs without losing too much information. Rather than building a GNN on the full training graph and then sampling across the layers, we first sample the training graph and partition it into several chordal subgraphs with balanced size. By doing so, CGST can significantly reduce the CPU memory required for GNN training, making it more scalable to larger graph datasets. Besides, CGST also applies random aggregation technique among subgraphs to alleviate the the influence of partitioning the whole graph. By building a complete GNN model on the subgraphs, CGST aims to maintain the performance characteristics of the full-batch training, avoiding the accuracy degradation seen in some prior sampling-based approaches. This allows us to capture the inherent node dependencies within the sampled subgraphs while avoiding the need to store the entire computation graph in GPU memory during back-propagation.

Our main contributions can be summarized as below:

- **Balanced Chordal Subgraph Partition Module.** To solve the first challenge that previous graph partition methods are difficult to form tight clusters, we build a graph partition module to split the original graph into subgraphs with balanced size. In this module, we extract appropriately connected subgraphs so that little information is lost when propagating within the subgraphs. A graph partition method is applied to generate several well-partitioned chordal subgraphs, which means chordal graphs with balanced sizes.

- **Random Aggregation Module.** To solve the second challenge that graph clustering algorithms tend to remove edges and cross-community nodes from the original datasets, we propose a random aggregation clustering approach to incorporate between-cluster links and reduce variance across batches.

- **Evaluation on three large-scale datasets.** Under extensive experiments on four real-world datasets, we show that CGST provides consistent boosts in the performance of node classification tasks over large-scale graph datasets.

## 2 RELATED WORKS

### 2.1 LAYER SAMPLING

A neural network model that extends convolution operation to the graph domain is first proposed by Bruna et al. (2013). Further, Kipf & Welling (2016); Defferrard et al. (2016) speed up graph convolution computation with localized filters based on Chebyshev expansion. They target relatively small datasets and thus the training proceeds in full batch. In order to scale GCNs to large graphs, layer sampling techniques (Hamilton et al., 2017; Chen et al., 2018b; Ying et al., 2018b; Chen et al., 2018a; Gao et al., 2018; Huang et al., 2018) have been proposed for efficient minibatch training. The layer sampling algorithm of GraphSAGE (Hamilton et al., 2017) performs uniform node sampling

on the previous layer neighbors. It enforces a pre-defined budget on the sample size, so as to bound the minibatch computation complexity.

S-GCN (Chen et al., 2018a) further restricts neighborhood size by requiring only two support nodes in the previous layer. The idea is to use the historical activations in the previous layer to avoid redundant re-evaluation. FastGCN (Chen et al., 2018b) performs sampling from another perspective. Instead of tracking down the inter-layer connections, node sampling is performed independently for each layer. It applies importance sampling to reduce variance, and results in constant sample size in all layers. However, the minibatches potentially become too sparse to achieve high accuracy. Huang et al. (2018) improves FastGCN by an additional sampling neural network. It ensures high accuracy, since sampling is conditioned on the selected nodes in the next layer. Significant overhead may be incurred due to the expensive sampling algorithm and the extra sampler parameters to be learned. In addition, the work in Zeng et al. (2018a) proposes a subgraph based training algorithm that is scalable with respect to GCN depth, and also highly parallelizable on multi-core machines.

## 2.2 SUBGRAPH SAMPLING

Instead of sampling layers, the works of Zeng et al. (2018b) and Chiang et al. (2019) build minibatches from subgraphs. Zeng et al. (2018b) proposes a specific graph sampling algorithm to ensure connectivity among minibatch nodes. They further present techniques to scale such training on shared-memory multi-core platforms. More recently, ClusterGCN (Chiang et al., 2019) proposes graph clustering based minibatch training. During pre-processing, the training graph is partitioned into densely connected clusters. During training, clusters are randomly selected to form minibatches, and intra-cluster edge connections remain unchanged.

## 2.3 GNN DECOUPLING

Another line of research focuses on improving model capacity. Applying attention on graphs, the architectures of Zeng et al. (2019) better capture neighbor features by dynamically adjusting edge weights.Klicpera et al. (2018) combines PageRank with GCNs to enable efficient information propagation from many hops away. To develop deeper models, "skip-connection" is borrowed from CNNs (He et al., 2015; Huang et al., 2017) into the GCN context. In particular, JK-net Xu et al. (2018) demonstrates significant accuracy improvement on GCNs with more than two layers. Note, however, that JK-net (Xu et al., 2018) follows the same sampling strategy as GraphSAGE (Hamilton et al., 2017). Thus, its training cost is high due to neighbor explosion. In addition, high order graph convolutional layers (Zhou, 2017; Lee et al., 2018; Abu-El-Haija et al., 2019) also help propagate long-distance features. With the numerous architectural variants developed, the question of how to train them efficiently via minibatches still remains to be answered.

# 3 PREMILINARY

## 3.1 PROBLEM DEFINITION

**Problem Definition 1** *Node Classification*.

*Given a graph $\mathcal{G}(V, E, W)$ with a subset of nodes $V_l \subset V$ labeled, where $V$ is the set of $n$ nodes in the graph (possibly augmented with other features), and $V_u = V \backslash V_l$ is the set of unlabeled nodes. Here $W$ is the weight matrix, and $E$ is the set of edges. Let $Y$ be the set of $m$ possible labels, and $Y_l = \{y_1, y_2, \ldots, y_l\}$ be the initial labels on nodes in the set $V_l$. The task is to infer labels $\widetilde{Y}$ on all nodes $V$ of the graph.*

*Let $V_l$ be the set of $l$ initially labeled nodes and $V_u$ be the set of $n - l$ unlabeled nodes such that $V = V_l \cup V_u$. We assume the nodes are ordered such that the first $l$ nodes are initially labeled and the remaining nodes are unlabeled so that $V = \{v_1, \ldots, v_l, v_{l+1}, \ldots, v_n\}$. An edge $(i, j) \in E$ between nodes $v_i$ and $v_j$ has weight $w_{ij}$.*

**Problem Definition 2** *Multi-class Classification*. *For multi-class classification, $y_i$ denotes a probability distribution over $Y$, where $Y$ is the set of possible labels. For any label $c \in Y$, $y_i[c]$ is the*

*probability of labeling node $v_i$ with label c. Here, $Y_l$ is a matrix of size $l \times m$. The output of the node classification problem is labels $\widetilde{Y}$ on all nodes in $V$.*

## 3.2 BACKGROUND

**Graph Neural Networks.** In this part, we introduce background on sampling-based training for GNNs to facilitate further discussion. A GNN learns a representation of an un-directed, attributed graph $G = (\mathcal{V}, \mathcal{E}, A)$, where each node $v \in \mathcal{V}$ has a length-$f$ attribute $x_v$. Suppose $A$ is the adjacency matrix and $\widetilde{A}$ is the normalized one (i.e., $\widetilde{A} = D^{-1}A$, and $D$ is the diagonal degree matrix), we have:

$$\widetilde{A} = A + I, \widetilde{D}_{ii} = \sum_j \widetilde{A}_{ij} \tag{1}$$

Suppose a $L$-layer GNN consists of $L$ graph convolution layers and each of them constructs embeddings for each node by mixing the embeddings of the node's neighbors in the graph from the previous layer:

$$Z^{(l+1)} = A'X^{(l)}W^{(l)} \tag{2}$$
$$X^{(l+1)} = \sigma(Z^{(l+1)}) \tag{3}$$

where $X^{(l)} \in \mathbb{R}^{N \times F_l}$ is the embedding at the $l$-th layer for all the $N$ nodes and $X^{(0)} = X$. $X^{l+1}$ is the embedding for $l+1$-layer. $A'$ is the normalized and regularized adjacency matrix and $W^{(l)} \in \mathbb{R}^{F_l \times F_{l+1}}$ is the feature transformation matrix that will be learned for the downstream tasks.

GNNs can be applied under inductive and transductive settings. In this paper, we focus only on inductive learning. It has been shown that inductive learning is especially challenging (Hamilton et al., 2017) — during training, neither attributes nor connections of the test nodes are present. Thus, an inductive model has to generalize to completely unseen graphs.

**Graph Notations.** In this part, we introduce some notations for further discussion. For a graph $G$, the vertex set is denoted by $\mathcal{V}(G)$ and the edge set is denoted by $\mathcal{E}(G)$. For an edge $uv \in \mathcal{E}(G)$, we call $u$ and $v$ its endpoints. We say that $G$ is isomorphic to $G$ if there is a bijection $\phi : \mathcal{V}(G) \longrightarrow \mathcal{V}(G)$ such that for all $u, v \in \mathcal{V}(G)$, $uv \in \mathcal{V}(G)$ if and only if $\phi(u)\phi(v) \in \mathcal{E}(G)$. We say that $G$ is a subgraph of $H$, denoted by $G \subseteq H$, if $\mathcal{V}(G) \subseteq \mathcal{V}(H)$ and $\mathcal{E}(G) \subseteq \mathcal{E}(H)$.

For graphs $G$ and $H$, let $G \cup H$ be the graph with vertex set $V(G) \cup V(H)$ and edge set $\mathcal{E}(G) \cup \mathcal{E}(H)$. For a vertex $v$ of a graph $G$, $N_G(v) := \{w \in \mathcal{V}(G) | vw \in \mathcal{E}(G)\}$ is the set of neighbors of $v$ in $G$. The degree of $v$ is $\deg_G(v) := |N_G(v)|$. Given a set $X \subseteq V(G)$, we define:

$$N_G(X) = \bigcup_{v \in X} N_G(v) \backslash X \tag{4}$$

$$N_G[X] = N_G(X) \cup X \tag{5}$$

## 4 METHODOLOGY

### 4.1 OVERVIEW

**Overall Architecture.** In order to maintain prior information of the original graph structure while processing mini-batch training, we propose a **C**hordal **G**raph **S**ampling-based mini-batch **T**raining algorithm for GNNs on large scale graph datasets, called **CGST**. CGST includes two modules: a balanced chordal graph partition module and a batch random aggregation module. The first module extracts appropriately connected subgraphs so that little information is lost when propagating within the subgraphs. A graph partition method is applied to generate several well-partitioned chordal subgraphs, which means chordal graphs with balanced sizes. The second module solves the challenge that graph clustering algorithms tend to remove edges and cross-community nodes from the original datasets. By incorporating these two modules, CGST can achieve better performance without increasing excessive training time and training costs. The overview of CGST is shown in Figure 2.

**Graph Partitioning.** For a given graph $\mathcal{G}$, we partition its nodes into $c$ groups: $\mathcal{V} = [\mathcal{V}_1, \cdots \mathcal{V}_c]$ where $\mathcal{V}_t$ consists of the nodes in the $t$-th partition. Thus we have $c$ subgraphs as

$$\bar{G} = [G_1, \cdots, G_c] = [\{\mathcal{V}_1, \mathcal{E}_1\}, \cdots, \{\mathcal{V}_c, \mathcal{E}_c\}] \tag{6}$$

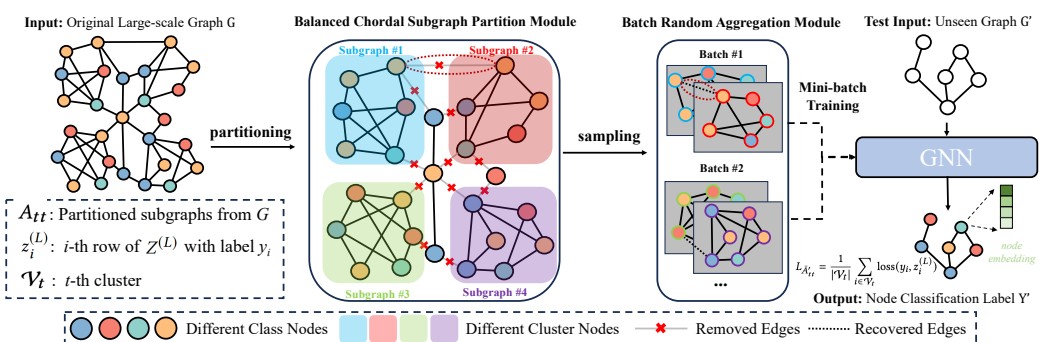

Figure 2: The overall architecture of CGST.

where each $\mathcal{E}_t$ only consists of the links between nodes in $\mathcal{V}_t$.

After reorganizing nodes, the adjacency matrix is partitioned into $c^2$ submatrices, where each diagonal block $A_{tt}$ is a $|\mathcal{V}_t| \times |\mathcal{V}_t|$ adjacency matrix containing the links within $G_t$. $\bar{A}$ is the adjacency matrix for graph $\bar{G}$; $A_{st}$ contains the links between two partitions $\mathcal{V}_s$ and $\mathcal{V}_t$. Similarly, we can partition the feature matrix $X$ and training labels $Y$ according to the partition $[\mathcal{V}_1, \cdots, \mathcal{V}_c]$ as $[X_1, \cdots, X_c]$ and $[Y_1, \cdots, Y_c]$ where $X_t$ and $Y_t$ consist of the features and labels for the nodes in $V_t$ respectively.

**Loss Function.** The benefit of this block-diagonal approximation $\bar{G}$ is that the objective function of GNN becomes decomposable into different batches (clusters). Let $\bar{A}'$ denotes the normalized version of $\bar{A}$, the final embedding matrix becomes

$$Z^{(L)} = \bar{A}'\sigma(\bar{A}'\sigma(\cdots\sigma(\bar{A}'XW^{(0)})W^{(1)})\cdots)W^{(L-1)} \tag{7}$$

due to the block-diagonal form of $\bar{A}$ (note that $\bar{A}'_{tt}$ is the corresponding diagonal block of $\bar{A}'$). The loss function can also be decomposed into

$$L_{\bar{A}'} = \sum_t \frac{|\mathcal{V}_t|}{N} L_{\bar{A}'_{tt}} \quad \text{and} \quad L_{\bar{A}'_{tt}} = \frac{1}{|\mathcal{V}_t|} \sum_{i \in \mathcal{V}_t} \text{loss}(y_i, z_i^{(L)}). \tag{8}$$

**Training Algorithm of CGST.** The training procedure of CGST follows the common rules of mini-batch training. At each step, we sample a cluster $\mathcal{V}_t$ and conduct SGD to update based on the gradient of $L_{\bar{A}'_{tt}}$, and this only requires the sub-graph $A_{tt}$, the $X_t$, $Y_t$ on the current batch and the models $\{W^{(l)}\}_{l=1}^L$. The implementation only requires forward and backward propagation of matrix products, which is much easier to implement than the neighborhood search procedure used in previous SGD-based training methods.

## 4.2 BALANCED CHORDAL GRAPH PARTITION

In this module, our goal is to extract appropriately connected subgraphs so that little information is lost when propagating within the subgraphs. We propose a graph partition method to generate several well-partitioned chordal subgraphs, which means chordal graphs with balanced sizes.

A chordal graph is one in which all cycles of four or more vertices have a chord, which is an edge that is not part of the cycle but connects two vertices of the cycle. Equivalently, every induced cycle in the graph should have exactly three vertices. The chordal graphs may also be characterized as graphs that have perfect elimination orderings, as graphs in which each minimal separator is a clique, and as the intersection graphs of subtrees of a tree. In mathematical, a chordal graph can be formally defined as below:

**Definition 1** *Chordal graph. A graph is chordal if every cycle of length at least 4 has a chord. A vertex $v$ of $G$ is called simplicial in $G$ if $N(v)$ is a clique in $G$. The ordering $\{v_1, \ldots, v_n\}$ of the vertices of $G$ is a perfect elimination order of $G$ if for all $i$, $v_i$ is simplicial in $G[v_1, \ldots, v_i]$. Also, a graph is chordal if it has a perfect elimination order.*

---

**Algorithm 1:** Find well-partitioned chordal subgraphs $G^c$ given the whole graph $G$ and a clique tree $\mathcal{T}_{K^c}$ of it.

---

**Input:** A clique tree $\mathcal{T}_{K^c}$ of a large graph $G$.
**Output:** Balanced partitions of $G$.
1: **Function** *FindBalancedPartition*( $\mathcal{T}_{K^c}$ )
2: **if** $|V(\mathcal{T}_{K^c})| \leq 1$ **then**
3:    **return** $\emptyset$
4: **end if**
5: Select an arbitrary edge $e$ to cut $\mathcal{T}_{K^c}$ into two components $\mathcal{T}_{K_1^c}$ and $\mathcal{T}_{K_2^c}$
6: $L = \bigcup_{K \in V(\mathcal{T}_{K_1^c})} K \setminus e$
7: $R = \bigcup_{K \in V(\mathcal{T}_{K_2^c})} K \setminus e$
8: **return** $\{\{L, R\}\} \cup$ *FindBalancedPartition*($\mathcal{T}_{K_1^c}$) $\cup$ *FindBalancedPartition*($\mathcal{T}_{K_2^c}$)
9: **End function**

---

Similarly, according to Ahn et al. (2022), the well-partitioned chordal graph can be formally defined as below:

**Definition 2** *Well-partitioned chordal graph*. *A graph is a well-partitioned chordal graph if and only if it has no induced subgraph isomorphic to a graph in $\mathcal{O}$. Furthermore, there is a polynomial-time algorithm that given a graph $G$, outputs either an induced subgraph of $G$ isomorphic to a graph in $\mathcal{O}$, or a partition tree of each connected component which confirms that $G$ is a well-partitioned chordal graph.*

As is proved in Ahn et al. (2022), a graph can be divided into several well-partitioned chordal subgraphs in polynomial time. It is easy to see that every well-partitioned chordal graph $G$ is a chordal graph because every leaf of the partition tree of a component of $G$ contains a simplicial vertex of $G$, and after removing this vertex, the remaining graph is still a well-partitioned chordal graph. Thus, we could construct a perfect elimination ordering to generate well-partitioned chordal subgraphs. In Summary, the algorithm of finding balanced chordal graph partitions is presented in Algorithm 1.

### 4.3 RANDOM AGGREGATION

In this module, in order to fetch each node $i$'s neighbor nodes' embeddings, we need to further aggregate each neighbor node's neighbor nodes' embeddings as well. Although vanilla subgraph-sampling based method achieves good computational and memory complexity, there are still two potential issues:

- After the graph is partitioned, some links are removed. Thus the performance could be affected.
- Graph clustering algorithms tend to bring similar nodes together. Hence the distribution of a cluster could be different from the original data set, leading to a biased estimation of the full gradient while performing SGD updates.

In order to tackle the aforementioned challenges, we introduce a batch random aggregation mechanism designed to integrate inter-cluster connections and reduce variability across batches. Initially, we segment the graph into clusters denoted as $k$ clusters $\mathcal{C}_1, \cdots, \mathcal{C}_k$ with a relatively large value of $k$. When forming a batch $B$ for a Stochastic Gradient Descent (SGD) update, instead of selecting a single cluster, we randomly pick $n$ clusters, denoted as $c_1, \ldots, c_n$ and include their nodes $\mathcal{V}_{c_1} \cup \cdots \cup \mathcal{V}_{c_n}$ into the batch. Moreover, the connections between the chosen clusters,

$$\{A_{ij} \mid i, j \in c_1, \ldots, c_n\}$$

are reintroduced. This method ensures the reintegration of inter-cluster links and reduces batch-to-batch variance through diverse cluster combinations. The training procedure of the Cluster Graph Spatial Transformer (CGST) is outlined in Algorithm 2. In each epoch, varied subgraph combinations are selected as batches. An experiment is performed on three extensive graph datasets to showcase the efficacy of the proposed approach. The results in Table 2 demonstrate that utilizing multiple clusters within a batch can enhance performance.

---

**Algorithm 2:** Training Algorithm of CGST

---

**Input:** Graph $\mathcal{G}$, feature $X$, label $Y$
**Output:** Predicted node label $\widetilde{Y}$
1: Partition graph into $c$ clusters $\mathcal{V}_1, \mathcal{V}_2, \cdots, \mathcal{V}_c$ with **Chordal Subgraph Partition Module**
2: **while** iter $<$ max_iter **do**
3:     Randomly choose $n$ clusters, $c_1, \cdots, c_n$
4:     Form the batch $\bar{B}$ with nodes $[N(\mathcal{V}_{t_1}), N(\mathcal{V}_{c_1}, \cdots, N(\mathcal{V}_{c_n})]$ and links $\mathcal{E}$
5:     Compute $g \leftarrow \nabla L_{A_{\bar{\mathcal{V}}, \bar{\mathcal{V}}}}$ (loss is introduced in Equation 8)
6:     Conduct Mini-batch SGD using gradient estimator $g$
7: **end while**
8: **return** $\{W_l\}_{l=1}^L$

---

## 5 EXPERIMENTS

### 5.1 EXPERIMENT SETUP

#### 5.1.1 DATASETS

We evaluate our model on three large-scale graph datasets through the node classification task. Some basic information of these three datasets is demonstrated as below:

- **Twibot-22** (Feng et al., 2022). Twibot-22 is a comprehensive graph-based Twitter bot detection benchmark that presents the largest dataset to date, provides diversified entities and relations on the Twitter network, and has considerably better annotation quality than existing datasets.

- **Hyperlink Graph** (Lehmberg et al., 2014). Hyperlink graphs have been extracted from the 2012 and 2014 versions of the Common Crawl web corpera. We use the 2014 graph, which covers 1.7 billion web pages connected by 64 billion hyperlinks.

- **MalNet** (Freitas et al., 2020). MalNet is a large public graph database, representing a large-scale ontology of software function call graphs. MalNet contains over 1.2 million graphs, averaging over 17k nodes and 39k edges per graph, across a hierarchy of 47 types and 696 families.

#### 5.1.2 BASELINE METHODS

We compare our model with several baseline methods for large-scale graph datasets, including sampling-based and decoupling-based methods. We select six baselines to evaluate the performance of our model on the node classification task. The basic information of these baseline methods is demonstrated below:

- **ClusterGCN** (Chiang et al., 2019). ClusterGCN first partitions the entire graph into clusters based on some graph partition algorithms, e.g. METIS (Karypis & Kumar, 1998), and then selects several clusters to form a batch.

- **GraphSAINT** (Zeng et al., 2019). GraphSAINT samples a subset of nodes based on a sampling strategy and then induces the corresponding subgraph as a batch. The commonly-used sampling strategies include: a node sampler: $(\mathbb{P}(u) = ||\widetilde{A}_{:,u}||^2)$, an edge sampler: $(\mathbb{P}(u, v) = \frac{1}{deg(u)} + \frac{1}{deg(v)})$, and a random walk sampler.

- **GnnAutoScale** (Fey et al., 2021). GAS incorporates historical embeddings to provably maintain the expressive power of full-batch GNN. It provides approximation error bounds of historical embeddings and show how to tighten them in practice.

- **SIGN** (Frasca et al., 2020). SIGN concatenates features from different hops and then fuse them as the final node representation via a linear layer.

- **SAGN**. SAGN adopts attention mechanism to combine feature representations from $K$ hops: $\bar{X} = \sum_{l=1}^K T^l X^l$, where $T^l$ is a diagonal matrix whose diagonal corresponds to the attention weight for each node of $k$-hop information.

### 5.1.3 EXPERIMENT SETTINGS

We implement our proposed method CGST in PyTorch. For the other methods, we use all the original papers' code from their Github pages. Since some baseline methods has difficulty scaling to large graphs, we do not compare with it here. For all the methods, we use the Adam optimizer with a learning rate as 0.01, a dropout rate as 20%, weight decay as zero. In each experiment, we consider the same GCN architecture for all methods. For SIGN and SAGN, we follow the settings provided by the original papers and set the batch sizes as 512. For our model, the clustering is seen as a preprocessing step and its running time is not taken into account in training time. All the experiments are conducted on four machines with two NVIDIA 3090 GPUs and 128 GB memory on Ubuntu 20.04. Codes are available at `https://anonymous.4open.science/r/CGST-6225/`

### 5.2 EXPERIMENT RESULTS

Table 1: Performance and efficiency comparison between CGST and other baseline methods on three large-scale graph datasets. Four metrics in terms of F1 score(%), accuracy(%), memory usage(MB) and training time are evaluated. All experiments are repeated three times. For F1 score and accuracy, the mean and standard deviation ($\pm$) are reported. For memory usage and training time, the average scores are reported. The best results are in bold and the second best results are underlined.

| Method | Twibot | | | | Hyperlink Graph | | | | MalNet | | | |
|---|---|---|---|---|---|---|---|---|---|---|---|---|
| | F1 Score | Acc | Mem Usage | Training Time | F1 Score | Acc | Mem Usage | Training Time | F1 Score | Acc | Mem Usage | Training Time |
| ClusterGCN | 58.62 ±1.25 | 90.30 ±0.89 | 2462 | 2.13h | 35.27 ±0.49 | 49.19 ±0.32 | 6591 | 3.7h | 41.45 ±2.60 | 45.28 ±1.87 | 6815 | 23.1h |
| GraphSAINT | 66.76 ±1.46 | 93.05 ±0.93 | 2026 | 2.6h | 37.46 ±0.14 | 56.64 ±0.10 | **4605** | **2.1h** | 50.47 ±0.60 | 53.00 ±0.87 | 8562 | 18.5h |
| GAS | 53.76 ±0.57 | 87.51 ±0.41 | **1010** | 3.5h | OOM | OOM | OOM | OOM | 20.90 ±0.36 | 26.79 ±0.32 | **4406** | 15.4h |
| SIGN | 53.07 ±1.03 | 83.62 ±0.82 | 4463 | 0.5h | OOM | OOM | OOM | OOM | 34.13 ±0.40 | 37.76 ±0.28 | 10979 | **10.4h** |
| SAGN | 52.58 ±0.85 | 84.46 ±0.74 | 4719 | **0.3h** | OOM | OOM | OOM | OOM | OOM | OOM | OOM | OOM |
| **CGST** | **66.91** ±**1.18** | **93.47** ±**0.77** | 1804 | 2.5h | **45.31** ±**0.35** | **57.18** ±**0.39** | 10499 | 3.3h | **53.20** ±**0.12** | **54.29** ±**0.07** | 10370 | 26.3h |

A comparison between our methods and other baseline methods on three large-scale graph datasets is shown in Table 1. We use four metrics evaluate the models from the perspective of performance and efficiency:

- **F1 score:** The macro F1-score of model evaluation on test data.

- **Accuracy:** The accuracy of model evaluation on test data.

- **Memory Usage:** Total memory costs of model parameters and all hidden representations when training a batch.

- **Training Time:** The total training time (exclude validation) before convergence point.

**Performance.** First we compare CGST with other methods in terms of F1 score and accuracy. As is shown in Table 1, our proposed CGST can achieve the highest F1 score and accuracy score among all the methods, without increasing excessive memory usage and training time. One surprising thing is that the subgraph sampling-based methods such as ClusterGCN and GraphSAINT can achieve higher accuracy than the decoupling-based methods. This is probably because the graph data is incomplete and noisy, and the stochastic nature of the sampling method can bring in regularization for training a more robust graph neural network with better generalization accuracy. Another observation is that no matter the size of the graph, CGST can still converge well, while some decoupling-based methods cannot scale to billions of nodes. This indicates that CGST is scalable to training very large GNN while maintaining high accuracy.

**Efficiency.** For training large-scale GNNs, besides performance, memory usage needed for training and training time are important and will directly restrict the scalability. The memory usage includes

the memory needed for training the GCN for many epochs. As discussed in Section 2, to speed up training, SIGN needs to save historical embeddings during training, so it needs much more memory for training than sampling-based methods. In Table 1, we can see that when maintaining more information to boost performance, CGST's memory usage and training time do not increase a lot. The reason is that the CGST The essence of preserving information is to retain the structural information of the original graph when generating a batch, which will not bring too much memory cost. Also, the more reasonable batch division proposed in Section 4.2 can also save CGST from excessive training time.

## 5.3 ABLATION STUDY

Table 2: Performance comparison between CGST and its two variants. Four metrics including F1 score, accuracy, memory usage, and training time are reported. All experiments are repeated three times. For F1 score and accuracy, the mean and standard deviation ($\pm$) are reported. For memory usage and training time, the average scores are reported.

| Method | Twibot | | | | Hyperlink Graph | | | | MalNet | | | |
|---|---|---|---|---|---|---|---|---|---|---|---|---|
| | F1 Score | Acc | Mem Usage | Training Time | F1 Score | Acc | Mem Usage | Training Time | F1 Score | Acc | Mem Usage | Training Time |
| CGST w/o CGPM | **68.04** $\pm$**0.61** | **93.89** $\pm$**0.64** | 2306 | 3.7h | OOM OOM | OOM OOM | OOM OOM | OOM OOM | OOM OOM | OOM OOM | OOM OOM | OOM OOM |
| CGST w/o BRAM | 62.92 $\pm$3.53 | 87.34 $\pm$4.88 | **1633** | 4.8h | 34.00 $\pm$0.73 | 51.08 $\pm$0.62 | **7551** | 5.3h | 51.46 $\pm$0.21 | 47.30 $\pm$0.10 | **8608** | 30.1h |
| **CGST** | 66.91 $\pm$1.18 | 93.47 $\pm$0.77 | 1804 | **2.5h** | **45.31** $\pm$**0.35** | **57.18** $\pm$**0.39** | 10499 | **3.3h** | **53.20** $\pm$**0.12** | **54.29** $\pm$**0.07** | 10370 | **26.3h** |

As discussed in Section 4, CGST includes two novel modules: a balanced chordal graph partition module (denoted as CGPM) and a batch random aggregation module (denoted as BRAM). We perform an ablation study to examine the effect of these two modules. To evaluate, we consider two variants of CGST:

1. CGST without chordal graph partition module (abbreviated as CGST w/o CGPM).

2. CGST without batch random aggregation module (abbreviated as w/o BRAM).

Ablation study results on three large-scale datasets are shown in Table 2, from which we could observe that these two modules improve CGST from two different perspectives. Table 2 reveals insights into the individual contributions of the CGPM and BRAM modules. The absence of the CGPM module leads to a noticeable decline in performance metrics, particularly in scenarios where graph structure is pivotal for model efficacy. Although the performance CGST w/o CGPM on a median scale dataset (e.g., Twibot) is slightly better than that of CGST due to the removal of the constraint on subgraph size, the unbalanced number of subgraphs greatly limits its scalability, making it unable to support node classification tasks on larger datasets like Hyperlink Graph and MalNet.

Conversely, omitting the BRAM module results in a distinct degradation in the model's ability to effectively aggregate information across batches, highlighting the module's role in enhancing information flow and convergence speed during training. In addition, we notice that although the memory usage of CGST w/o BRAM is slightly reduced compared to the CGST, its training time on all three datasets is longer. This is because chordal graph partitioning algorithm tend to bring similar nodes together. Hence the distribution of a cluster could be different from the original data set, leading to a biased estimation of the full gradient while performing SGD updates. In summary, the experiment results in Table 2 underscore the indispensable roles played by the CGPM and BRAM modules in bolstering the overall performance and efficacy of CGST.

## 5.4 PARAMETER SENSITIVITY ANALYSIS

There is one important hyper-parameters that should be conducted, which is the ratio of maximum number of nodes in a subgraph to the whole graph (denoted as $\alpha$). We test the sensitivity of $\alpha$ using the regular experiment setting on Twibot and Hyperlink Graph dataset. We vary the $\alpha$ from

Table 3: Clustering performance comparison between chordal graph partition method and two other partition methods. Two metrics including NMI and the percentage of removed edges are reported.

| Method | Twibot | | Hyperlink Graph | | MalNet | |
|---|---|---|---|---|---|---|
| | NMI | Removed Edges | NMI | Removed Edges | NMI | Removed Edges |
| Random Partition | 0.31 | 21.46% | 0.18 | 13.52% | 0.20 | 20.08% |
| METIS | 0.53 | 9.21% | 0.41 | 5.80% | **0.36** | **8.54%** |
| Balanced Chordal Graph Partition | **0.72** | **5.53%** | **0.44** | **5.03%** | 0.28 | 11.07% |

$\{0.01, 0.02, 0.05\}$, $\{1e^{-5}, 2e^{-5}, 5e^{-5}\}$ on the Twibot dataset and Hyperlink graph dataset respectively. The corresponding F1 score and training timer are respectively shown in Figure 3 with blue lines. The blue shaded area in each line chart indicates the error range of the corresponding standard deviation. From the line chart in Figure 3, without affecting the scalability of CGST, $\alpha$ is not sensitive over these two datasets. In summary, the selection of $\alpha$ is a trade-off between performance and efficiency. A larger $\alpha$ allows CGST to retain more original graph structure information during mini-batch training and then boost performance, but it will also increase training time due to batch imbalance.

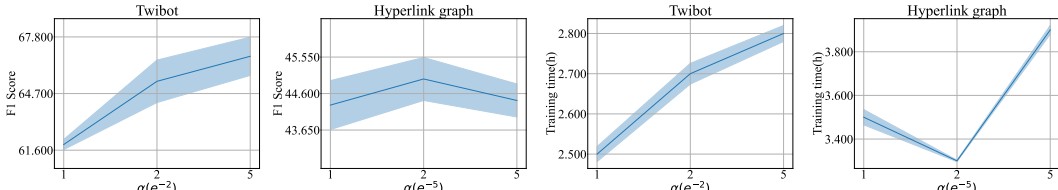

Figure 3: Parameter Sensitivity analysis of $\alpha$ over Twibot dataset and Hyperlink graph dataset. The corresponding F1 scores and training time are respectively shown using line charts. The blue shaded area in each line chart indicates the error range of the corresponding standard deviation.

## 5.5 CASE STUDY

To further substantiate the effectiveness of the chordal graph partitioning algorithm introduced in Section 4, a comprehensive evaluation was conducted across three distinct datasets. The efficacy of our method was rigorously scrutinized through a comparative analysis against established techniques such as METIS and random partitioning. By employing key evaluation metrics including NMI (Normalized Mutual Information) and the quantification of removed edges, a thorough assessment of the clustering quality was achieved. The experiment results from Table 3 affirm the superiority of our proposed approach, underscoring its capability to consistently generate higher-quality clusters in diverse graph partitioning scenarios.

## 6 CONCLUSION

We propose a new algorithm namely CGST for scaling GNNs to large-scale graph datasets. CGST includes two modules: a balanced chordal graph partition module and a batch random aggregation module. The first module extracts appropriately connected subgraphs so that little information is lost when propagating within the subgraphs. A graph partition method is applied to generate several well-partitioned chordal subgraphs, which means chordal graphs with balanced sizes. The second module solves the challenge that graph clustering algorithms tend to remove edges and cross-community nodes from the original datasets. Finally, under extensive experiments on four real-world datasets, we show that CGST provides consistent boosts in the performance of node classification tasks over large-scale graph datasets.

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
