# OpenReview forum: "Chordal Graph Sampling-Based Mini-batch Training Algorithm for Large Graphs"
_ICLR.cc/2025/Conference — ICLR 2025 Conference Withdrawn Submission_

### Official Review · Reviewer_XWcU · 2024-10-25

**Soundness:** 2
**Presentation:** 1
**Contribution:** 1
**Rating:** 3
**Confidence:** 3

**Summary:**

This paper proposes a GNN training technique based on subgraph sampling, which is based on chordal subgraph partition. The authors tested the performance of CGST training on GCN across three large graphs.

**Strengths:**

1. Experiments are conducted on three new datasets, which is inconsistent with previous work. Testing on new datasets is commendable.

2. This paper is easy to understand.

**Weaknesses:**

1. In the introduction, the figure 1 is inappropriate. Methods like CLUSTER-GCN and GAS use METIS to partition graphs, which does not result in some nodes being removed and unable to appear in the training batches.
2. The author frequently mentions that chordal subgraph partition is a major contribution, but I notice that the work in section 4.2 originates from [1]. This significantly undermines the novelty and amount of work in this paper. The author should provide an accurate explanation and description of this.
3. There are significant problems in the experimental section of the paper, which completely fails to meet the acceptance standards of ICLR. The author should provide experimental results for a variety of GNNs, not just limit to GCN. In terms of experimental results, CGST is also not ideal in terms of Mem usage and training time.  Moreover, the author should provide experimental results for commonly used datasets, such as Products.

Overall, I think this paper has significant deficiencies, especially in the experimental section.

[1]  Jungho Ahn,Lars Jaffke,O-joung Kwon, and Paloma TLima. Well-partitionedchordalgraphs. Discrete Mathematics.

**Questions:**

See weakness

---

### Official Review · Reviewer_eqDf · 2024-10-28

**Soundness:** 1
**Presentation:** 1
**Contribution:** 1
**Rating:** 3
**Confidence:** 3

**Summary:**

This paper proposes CGST. CGST includes a balanced chordal graph partition module and a batch random aggregation module to improve performance on node classification tasks while maintaining main information.

**Strengths:**

S1: The scalability of GNNs is an important research problem.

S2: The format looks fine.

**Weaknesses:**

W1: Apart from the graph partition method, I don't see any difference between this paper and ClusterGCN. And the graph partition method is adopted from existing work.

W2: The experiment setting is strange. The authors do not use common graph datasets. The results are also unsatisfied, as a scalable method, CGST performs poorly both in terms of memory and time.

W3: Parts of this paper were written in LLM, e.g., Line 321, what is "cluster graph spatial transformer"?

**Questions:**

Q1: Please discuss the difference between your paper and ClusterGCN.

Q2: In Line 94, "Under extensive experiments on four real-world datasets..." , where are the fourth dataset?

Q3: Section 2.3 has a title of "GNN decoupling", but the main text is about attention and skip-connection. How these concepts are related to GNN decoupling?

Q4: In Line 359, "We select six baselines to evaluate the performance...", where are the sixth baseline?

Q5: In Line 388, "Codes are available at...", there is no implementation code in this link, here is the tex source. It is normal not to provide the code during the review stage, but please do not deceive the reviewers.

Q6: In Line 517, "Case study...", this is not case study.

---

### Official Review · Reviewer_jFzN · 2024-10-30

**Soundness:** 2
**Presentation:** 3
**Contribution:** 2
**Rating:** 3
**Confidence:** 4

**Summary:**

This paper focuses on training GNNs on large graphs, and proposes to separate the whole graph into several balanced chordal graph. The authors try to maintain main information of the original graph structure.

**Strengths:**

1. The organization is good to follow.
2. The authors find a new potential of training large-scale graph.

**Weaknesses:**

1. Potential violation of double-blind policy. For the link at line 387, 'LICENSE' contains "Copyright (c) 2024 Su ziyang", where "Su ziyang" is a name implying one of the authors. Besides, this link contains the LATEX file of the submission not the code.
2. Section 3 is about one page, but only contains some well-known knowledge.
3. As shown in Figure 1, the authors argue that previous mini-batch methods suffer from information loss because of removed nodes and edges. However as shown in Figure 2, the proposed model also does not consider the nodes between different cliques.
4. The baselines are too old, where the authors do not provide the citation for SAGN.
5. As shown in Table 1, the proposed model cannot achieve the best memory usage and training time in all three datasets. Considering this paper studies large-scale training, these two metrics are very important.
6. I strongly suggest the authors further check the writing:
- Section 3 "PREMILINARY"
- What's $\mathcal{O}$ in Definition 2.

**Questions:**

1. In Algorithm 1, how to get the input clique tree? What's the complexity to construct such a tree? Considering to arbitrarily select an edge at each epoch, how to guarantee balanced partition?
2. What are the strengths to partition a graph into balanced chordal graph over other balanced partition?

---

### Official Review · Reviewer_Nmtu · 2024-11-06

**Soundness:** 2
**Presentation:** 2
**Contribution:** 1
**Rating:** 3
**Confidence:** 4

**Summary:**

This paper builds on cluster-gcn, using chordal graph partition instead of the metis algorithm. The performance of CGST was tested on three large-scale datasets. Overall, the novelty of this paper is limited, and its performance is relatively average.

**Strengths:**

1. This paper is easy to understand.

2. Experiments are conducted on three new datasets.

**Weaknesses:**

1. The novelty of this paper is extremely limited. The main difference from Cluster-GCN is the use of a different graph partition algorithm. Moreover, the graph partition algorithm used in this paper is not original. Additionally, the random aggregation technique mentioned in section 4.3 is also used by Cluster-GCN. The only difference is that edges between different clusters have been removed.
2. The experimental results indicate that CGST's performance is suboptimal. Although the accuracy is sufficiently good, as a work on scalable training, the memory usage and training time performance are worse than the baselines.
3. This paper does not discuss any work related to scalable training from 2022 to 2024.
4. This paper contains many typos.
5. This paper does not compare with baselines on standard datasets.

**Questions:**

See weakness

---

### Note · Authors · 2024-11-14

I have read and agree with the venue's withdrawal policy on behalf of myself and my co-authors.